Analysis and pollution evaluation of heavy metal content in soil of the Yellow River Wetland Reserve in Henan

Chen Xiaolong 1 2
Wong Cora Un In 1
Zhang Hongfeng 1 hfengzhang@mpu.edu.mo
1 Faculty of Humanities and Social Sciences, Macao Polytechnic University , Macao , China
2 Department of Management, Henan Institute of Technology , Xinxiang , China
Mahmood Haider
Electronic publication date: 2023 Dec 14
Publication date: 2023
Volume: 11
Electronic Location ID: e16454
Received 2023 Aug 25; Accepted 2023 Oct 23
Copyright: © 2023 Chen et al.
Copyright year: 2023
Copyright holder: Chen et al.
License: This is an open access article distributed under the terms of the Creative Commons Attribution License, which permits unrestricted use, distribution, reproduction and adaptation in any medium and for any purpose provided that it is properly attributed. For attribution, the original author(s), title, publication source (PeerJ) and either DOI or URL of the article must be cited.
License URL: https://creativecommons.org/licenses/by/4.0/

Keywords: Soil, Heavy metal, Pollution, Evaluation

Funding: The authors received no funding for this work.

==============================
Objective

This study aims to assess the contamination levels of six heavy metals, namely arsenic (As), cadmium (Cd), chromium (Cr), copper (Cu), mercury (Hg), and lead (Pb), in the soil of the Henan Yellow River Wetland Reserve. It seeks to reveal the spatial distribution and trends of heavy metal pollution, providing a scientific basis for the rational utilization and effective protection of soil. Additionally, it aims to propose targeted management and remediation recommendations to mitigate or prevent soil pollution.

Method

A total of 706 soil samples were collected in this area in combination with the land use type map. As and Hg were determined by atomic fluorescence spectrometry, and Cr, Cu, Pb and Cd were determined by inductively coupled plasma mass spectrometry. Taking the soil pollution risk screening value of agricultural land (GB15618-2018) as a reference value, the sample data were statistically analyzed, and the Nemerow comprehensive pollution index method combined with ArcGIS technology was used to evaluate the soil environmental quality.

Result

The comprehensive pollution index of the soil in the Yellow River Wetland Reserve was 0.42, ranging from 0.17 to 2.38, which was safe and not polluted (I grade). Out of 706 sampling locations, 674 remained uncontaminated, while 26 exhibited cleanliness. Although they were in the warning line, they did not exceed the standard, accounting for 3.68% of the total number of sampling points. Five sample points were slightly polluted, accounting for 0.71% of the total sample points, and one sample point was moderately polluted, accounting for 0.14% of the total sample points. It can be seen that there are few agricultural land pollution points in the Yellow River Wetland Reserve, and the soil environment quality is generally good.

Conclusion

The soil in the Yellow River Wetland Reserve in Henan has a very small amount of mild and moderate pollution, and there is no severe pollution. The cleanliness is currently high.

Introduction

Soil serves as a critical foundation for human survival and advancement, with its environmental quality and safety bearing profound importance for plant growth, human health, and even national ecological security (Zhang, Wu & Simonnot, 2012). The research on heavy metals in wetland soil is very important for the protection of its ecosystem. Wetlands are fragile due to the sensitive ecological environment, and it is difficult to recover after damage.

Concomitant with China’s brisk economic and social development, particularly the breakneck urbanization and industrialization, coupled with the widespread utilization of agricultural chemicals including pesticides, fertilizers and mulch, soil pollution has grown into an increasingly grave concern (Joshirvani, Samarghandi & Leili, 2021; Liu et al., 2021). Soil pollutants encompass a multitude of categories, including heavy metal, organic, biological and radioactive contaminations. The co-occurrence of diverse pollutant types further aggravates soil pollution to an alarming extent. Heavy metals are not easy to observe with the naked eye, exist for a long time, cannot be completely absorbed by plants, and cannot be degraded by soil microorganisms. They can also accumulate gradually through food chains and food webs, endangering human health. Therefore, in the contemporary era of building an environmentally friendly society, The pollution of soil heavy metals has aroused great attention (Dugassa, Diba & Bachie, 2021; Lin et al., 2020).

Previous research has mainly focused on the study and analysis of agricultural land in Henan coal mining areas (Li et al., 2017), shallow groundwater in Henan Province (Yang et al., 2022), soil in tobacco-growing areas of Henan (Chen et al., 2023) and agricultural soil heavy metal elements in Sanmenxia City (Cheng et al., 2015). These studies primarily emphasized heavy metals and nitrogen and phosphorus elements, with limited research on heavy metals such as lead, cadmium, and mercury in wetland conservation areas. Previous studies may not have comprehensively covered the entire wetland conservation area, resulting in gaps in our understanding of the spatial distribution of heavy metals, which is crucial for targeted management.

This study adopts the Nemerow comprehensive pollution index method and combines ArcGIS technology to assess the comprehensive pollution effects of various pollutants on the soil. The aim is to understand the development trends, key pollution areas, and the extent of composite pollution of six heavy metals. This research provides a foundation for the rational use and protection of soil, with the goal of preventing potential issues, protecting clean soil, and regulating and managing polluted soil.

Literature review

Literature review

Soil heavy metal pollution is a global environmental problem, posing potential threats to ecosystems and human health. In the past few decades, many foreign studies have focused on the source, migration, accumulation and ecological risk of heavy metals in soil.

Experts have carried out relevant research on the source and migration of heavy metals (Shi et al., 2022; Tu et al., 2023), and the research shows that heavy metals in soil mainly come from natural geological processes, industrial emissions, agricultural activities and urbanization. These heavy metals accumulate continuously in the soil through deposition, migration and transformation. The study also identified atmospheric deposition, water inflow, and plant uptake as principal transport avenues for heavy metals in soil. In terms of the ecological risk and impact of heavy metals in soil, relevant scholars have made relevant contributions (Cheng et al., 2010; Islam et al., 2019), and studies have shown that the accumulation of heavy metals in soil may lead to the decline of soil quality, affecting plant growth and crop yield. In addition, there has also been corresponding progress in soil food safety (Mohod & Dhote, 2013; Yi, Yang & Zhang, 2011), heavy metals in soil may also enter human body through the food chain, causing potential harm to health, such as poisoning and chronic diseases. Many studies have assessed the impact of different heavy metals on soil ecosystems and human health, as well as related risk assessment methods.

In order to deal with soil heavy metal pollution, researchers have proposed a variety of governance and restoration methods (Liu et al., 2018; Sun & Ottosen, 2012), such as soil remediation technology, phytoremediation, and bioremediation. The effectiveness and feasibility of these methods have been verified in different geographical environments. At the same time, the government and environmental protection agencies are also formulating policies and standards to control and reduce heavy metal pollution.

Emerging research avenues involve heavy metal transformations mediated by microorganisms, nanomaterial applications in remediation, and interactions between heavy metals and soil organic matter. Interdisciplinary efforts additionally probe the social and economic imprint of heavy metal pollution, alongside facets including social participation and risk communication.

Literature comments

Through the analysis of many foreign documents, we found a series of widely used research methods, including field investigation, laboratory analysis and simulation models. Among them, with the advancement of technology, molecular ecology and isotope tracing techniques have played an important role in revealing the biogeochemical cycle of heavy metals. However, some studies still have certain limitations in the choice of methods and design, such as insufficient sample size or limited geographical scope. Further studies are still needed to delve into the behavior and interactions of different heavy metals, as well as more effective remediation and remediation methods. These studies will help protect the ecological environment and human health.

Materials and Methods

Overview of the study area

The study area was the Henan Yellow River Wetland Reserve (112°21′49″–112°48′23″E; 34°33′59″–35°05′01″N), with a length of 301 km from east to west and a span of 50 km from north to south, according to the shape of the Yellow River and a banded distribution (Fig. 1).

Figure 1 Location distribution map of the Henan Yellow River Wetland Reserve.

The Henan Yellow River Wetland National Nature Reserve is a nature reserve approved by the State Council in June 2003 to protect wetland ecosystems and wetland waterfowl (Li et al., 2012; Qin et al., 2011). The protected area spans the four cities of Sanmenxia, Luoyang, Jiyuan and Jiaozuo, with an area of 680 km2. It belongs to the middle part of the alluvial plain of the Yellow River, and the soil is mostly sandy loamy fluvo-aquic soil. It belongs to the temperate monsoon climate, with an average annual precipitation of 726.5 mm and an annual average temperature of 14.1 °C. The Yellow River wetland in Henan is a place for many wild animals to forage, rest and overwinter. The wetland is rich in animal and plant resources and the original ecosystem is relatively complete.

Data source

The land use data of the Henan Yellow River Wetland Reserve are based on Google Earth Engine, and are preprocessed by mosaic, cloud removal and clipping. According to topographical and environmental factors, following the principles of representativeness, concentration and typicality of sample points, 706 sampling units were laid out in a grid (Fig. 2). When selecting sampling points in the Yellow River Wetland Reserve, we gave full consideration to achieving uniform distribution, geographical location, land use, and proximate pollution sources (e.g., nearby farmland or industrial areas). Sampling points were evenly distributed to preclude clustering in specific areas. We considered geographical factors including topography, landforms, and distribution of water bodies, which helped capture potential soil variations and gradients. Environmental factors such as human activities and pollution sources were considered to evaluate their impacts on soil quality. By carefully weighing these factors, representative sampling sites were selected to provide a more comprehensive understanding of the condition and distribution of soils in the Yellow River Wetland Reserve, laying a theoretical foundation for subsequent research.

Figure 2 Distribution map of sampling points in the Henan Yellow River Wetland Reserve.

The soil in the 0–20 cm plow layer was collected without touching the metal sampler. According to the principle of random, equal, and multi-point mixing, about 1 kg was taken according to the quartering method. It was put it in a cool place to air-dry naturally, and passed through a 0.15 mm mesh nylon sieve after air-drying and grinding.

Data processing

Excel 2010, DPS 7.65, and ArcGIS 10.2 were used for data processing. The data were processed using the principle of three times the standard deviation, and the measured value with a deviation of more than three times the standard deviation from the average value was considered an outlier. The outliers were verified and some outliers were eliminated after field inspection, and effective outliers were retained.

Evaluation method

Single factor pollution index

The single factor pollution index (Pi) is a method commonly used in China to evaluate heavy metal pollution, and its calculation formula is:

Pi=Ci/Cbi

Pi is the pollution index of heavy metal element i in soil, Ci is the measured content of heavy metal element i, and Cb is the soil pollution risk screening value of agricultural land for heavy metal element i (GB15618-2018) (Gao et al., 2021; Wang et al., 2019). When Pi ≤ 1, it means no pollution; when 1 < Pi ≤ 2, it means light pollution; when 2 < Pi ≤ 3, it means moderate pollution; when Pi > 3, it means heavy pollution.

Nemerow comprehensive pollution index method

The Nemerow comprehensive pollution index (comprehensive pollution index) is developed from the single-factor pollution index method (Kowalska et al., 2018; Rajkumar, Naik & Rishi, 2022; Sun & Ottosen, 2012). It is one of the most commonly used methods for comprehensive pollution index calculation at home and abroad. The weighted multi-factor environmental quality index can comprehensively reflect the pollution degree of different pollutants in the soil (Table 1). The specific calculation formula is:

Pn= (Pi(max)2+Pi(average)2)2

Pi(max) represents the maximum value of the single factor pollution index of each heavy metal element. Pi(average) represents the average value of the single factor pollution index of each heavy metal element. Pn represents the soil Nemerow comprehensive pollution index.

Table 1 Evaluation grade of soil pollution by the Nemerow comprehensive pollution index method.

Grade	Comprehensive pollution index (Pn)	Pollution level	Pollution description	
I	Pn ≤ 0.7	Clean	No pollution (clean)	
II	0.7 < Pn ≤ 1.0	Still clean	Not yet polluted (still clean)	
III	1.0 < Pn ≤ 2.0	(Warning line)
light pollution	Soil crops are all slightly polluted	
IV	2.0 < Pn ≤ 3.0	Moderately polluted	Soil crops are moderately polluted	
V	Pn > 3.0	Heavy pollution	Soil crops are polluted quite seriously	

ArcGIS technology

This study adopts the inverse distance weighted interpolation method in ArcGIS technology. The more sampling points, the more uniform the sampling points, and the more accurate the response to the pollution situation in the local area.

Results and analysis

Statistical analysis of test results

The coefficient of variation reflects the degree of dispersion of the data, and the total amount of the six heavy metals shows a certain degree of variability (Table 2). Large, extremely unevenly distributed, with areas of high content. From the perspective of kurtosis and skewness, the six heavy metals and pH presented a skewed distribution, arsenic (As), cadmium (Cd), copper (Cu), mercury (Hg), and lead (Pb) presented a right-skewed distribution, and pH and chromium (Cr) presented a left-skewed distribution. A total of 91% of the soil in the study area is alkaline, and the neutral and slightly acidic soils only account for 9%. Cd and Pb have obvious high value points, but As, Hg, Cr and Cu have no obvious high value points, indicating that soil Cd and Pb in the Yellow River Wetland Reserve should be paid attention to and tracked and monitored.

Table 2 Heavy metal content in the soil of the Yellow River Wetland Reserve.

Project	Range value	Average	Median value	Standard deviation	Coefficient of variation	Kurtosis	Skewness	
pH	5.16–9.03	–	8.30	0.59	7.30	9.49	−2.99	
As	2.86–21.8	9.46	9.40	2.62	27.67	2.54	0.96	
Cd	0.07–0.95	0.30	0.29	0,09	31.39	10.79	2.38	
Cr	34.2–102	63.8	64.4	9.39	14.71	0.39	−0.01	
Cu	9.99–49.3	25.8	25.8	5.51	21.37	0.94	0.36	
Hg	0.002–0.52	0.068	0.062	0.04	63.35	23.26	3.40	
Pb	10.8–554	26.6	24.9	22.71	85.37	429.94	19.46	
Note:

A dash (–) indicates that the pH value is not averaged.

Correlation analysis among heavy metal elements

It can be seen from Table 3 that there is a correlation between heavy metal elements, and the sources of indirect reaction heavy metals may be the same, and are greatly affected by the same pollution source (Song et al., 2022; Zhang et al., 2022). The data demonstrates a degree of homology in the soil elemental composition within the Yellow River Wetland Reserve. Correlation analysis and factor analysis showed that the soil heavy metal pollution in the Yellow River Wetland Reserve mainly came from the input of human activities, such as industrial pollution, ship sewage, breeding sewage, domestic sewage and agricultural non-point source pollution, followed by the input of natural factors.

Table 3 Correlation analysis of soil heavy metals in the Yellow River Wetland Reserve.

Project	As	Cd	Cr	Cu	Hg	Pb	
As	1						
Cd	0.03	1					
Cr	0.54**	0.03**	1				
Cu	0.49**	0.20**	0.60**	1			
Hg	0.15**	0.24**	−0.10	0.15**	1		
Pb	0.09**	0.83**	0.10**	0.21**	0.21**	1	
Notes:

* indicates p < 0.05, the lowest level of significance.

** indicates p < 0.01, the moderate level of significance.

The correlation coefficient of Pb and Cd is 0.83, showing a high correlation at the 0.01 level, indicating that the pollution sources of these two elements are the same; the correlation coefficients of Cr and As, Cu and As, Cu and Cr are 0.54, 0.49, 0.60, showing a 0.01 level The moderate correlation on, indicating that the sources of pollution between the above two elements are the same. Cr and Cd, Cr and Hg, As and Cd are not related, and the sources vary greatly. It can be seen that there is compound pollution between Pb and Cd, Cr and As, Cu and As, and Cu and Cr. According to the land use type and combined with the pollution caused by human activities such as forestry planting, scenic spots, and industry in the protected area (Barsova et al., 2019; Ottesen et al., 2008), It is necessary to further analyze the accumulation of pollution and control it at the source.

Evaluation of soil heavy metal single factor pollution index

Through the determination of the heavy metal content in the river basin soil of the Yellow River Wetland Reserve, the single-factor pollution of the pollution degree of Pb, Hg, Cd, As, Cu, and Cr was carried out using the soil pollution risk screening value of agricultural land (GB15618-2018) as the evaluation standard (Table 4).

Table 4 Single factor pollution index of heavy metals in soil of the Yellow River Wetland Reserve.

Pollution evaluation	As	Cd	Cr	Cu	Hg	Pb	
Single factor pollution index	0.520	0.441	0.326	0.363	0.032	0.730	
Individual pollution level	No pollution	No pollution	No pollution	No pollution	No pollution	No pollution	
Pollution level	I grade no pollution (clean)	

The results showed that the single factor pollution indices of As and Pb were higher, 0.520 and 0.730, respectively, and the single factor pollution indices of Cr and Hg were lower. The single-factor pollution index Pi values of heavy metals Pb, Hg, Cd, As, Cu, and Cr in the plow layer soil of farmland were all less than 1, and the exceeding rate was 0%.

In general, when the soil heavy metal single-factor pollution index was used to evaluate, the single-factor pollution index of Pb, Hg, Cd, As, Cu, Cr in the Yellow River Wetland Reserve was less than 1, and the single pollution level of the six heavy metal indicators was I grade (no pollution).

Evaluation results of Nemerow comprehensive pollution index method

Grading of comprehensive pollution index

The Nemerow comprehensive pollution index of the soil in the Yellow River Wetland Reserve is 0.42, ranging from 0.17 to 2.38, which is at the level I level of safety and pollution, and there is no risk of pollution (Table 5).

Table 5 Nemerow comprehensive pollution index of heavy metals in soil of the Yellow River Wetland Reserve.

Comprehensive pollution index (Pn)	Pn ≤ 0.7	0.7 < Pn ≤ 1.0	1.0 < Pn ≤ 2.0	2.0 < Pn ≤ 3.0	Pn > 3.0	Pn ≤ 0.7	
Pollution level	Clean	Still clean
(Warning Line)	Light pollution	Moderately
polluted	Heavy pollution	Clean	
Classification	I	II	I	III	IV	V	
Range value	0.17–2.38	
Average value
(standard deviation)	0.42 (0.15)	
Nemerow composite pollution index	0.42	
Pollution level	I grade no pollution (clean)	
Note:

The brackets in the distribution frequency of sampling points represent the percentage of the total number of sampling points.

Among the 706 sample points, 674 sample points were clean; 26 sample points were still clean, although they were in the warning line but did not exceed the standard, accounting for 3.68% of the total sample points. Five sample points were slightly polluted, accounting for 0.71% of the total sample points, and one sample point was moderately polluted, accounting for 0.14% of the total sample points (Table 6). It can be seen that there are few agricultural land pollution points in the Yellow River Wetland Reserve, and the soil environment quality is generally good.

Table 6 Distribution of soil heavy metal samples in the Yellow River Wetland Reserve.

Range value	Average value (standard deviation)	Sample point distribution frequency	
Pn ≤ 0.7	0.7 < Pn ≤ 1.0	1.0 < Pn ≤ 2.0	2.0 < Pn ≤ 3.0	Pn > 3.0	
Clean	Still clean (Warning line)	Light pollution	Moderately polluted	Heavy pollution	
I	II	I	III	IV	
0.17–2.38	0.42 (0.15)	674 (95.46)	26 (3.68)	5 (0.71)	1 (0.14)	0	

Combination of comprehensive pollution index

ArsGIS technology enabled normal distribution QQPlot, trend analysis, and statistical profiling of the pollution distribution per the comprehensive contamination index. It can be seen from Fig. 3 that the data set of the comprehensive pollution index falls on the reference line without deviation, showing a normal distribution.

Figure 3 Normal distribution of pollution index in quantile-quantile (Q-Q) plot.

It can be seen from Fig. 4 that geographically, there is a small amount of point-like pollution in the Yellow River Wetland Reserve, which is distributed in the north in a patchy form, but the patch area is small. Distributed in Mianchi, Xin’an County and Lakeside in the middle, they are still clean (warning line). Distributed in Lingbao in the east and Jiyuan City, Mengzhou City and Mengjin in the west. Mild pollution afflicted regions including the Sanmenxia Reservoir, Xiaolangdi Reservoir, Xixiayuan Off-Season Water Diversion Project, and the Yellow River River Service Control and Flood Control Project.

Figure 4 Trend of pollution index.

Comprehensive pollution area statistics

The grid area statistics (Table 7) by ArsGIS technology shows that the soil clean area of the Yellow River Wetland Reserve is 674.98 km2 (total area 680 km2), accounting for 99.26% of the total area. The clean (warning line) area is 5.141 km2, accounting for 0.75% of the total area. The lightly polluted area is 0.70 km2, and the lightly polluted area is 0.204 km2, accounting for 0.03% of the total area. There is no moderate and severe pollution. The limited extent of soil heavy metal contamination is evident in the Yellow River Wetland Reserve.

Table 7 Statistics on the comprehensive pollution area of the Yellow River Wetland Reserve.

Grade	Clean	Still clean (Warning line)	Light pollution	Moderately
polluted	Heavy pollution	
Area	674.98	16.41	0.70	0	0	
Proportion	0.09**	0.75	0.03	0	0	
Notes:

* indicates p < 0.05, the lowest level of significance.

** indicates p < 0.01, the moderate level of significance.

Discussion

Per the Nemerow comprehensive pollution index, soils in the Yellow River Wetland Reserve exhibit high cleanliness, with minor zones bordering warning thresholds and very few areas facing mild or moderate contamination. The heavy metal pollution likely stems from:

(1) There are small and medium-sized industrial and mining industries around lightly polluted areas. The high value point of Pb is near the Sanmenxia section of the Yellow River Wetland Reserve, and the hazardous waste disposal site, landfill site, industrial cluster area, mining area and other areas have a greater impact. Industrial areas are mainly polluted by oxidized Pb, plastic additives (including Pb), and heat stabilizers (including Pb). Industrial “three wastes” have become the main source of Pb pollution in this area.

The accumulation of heavy metals in soils can have detrimental impacts on both the environment and human health. Excessive amounts of heavy metals can impair soil microbial activity and nutrient cycling, reduce crop yields, and contaminate groundwater supplies. Heavy metal uptake in plants disrupts physiological processes and inhibits growth. In humans, chronic exposure to heavy metals through ingestion or inhalation can damage organs such as the kidneys, liver, and lungs. Heavy metals like lead, mercury, arsenic, and cadmium are particularly hazardous, even in low doses, and can cause developmental disorders, neurological damage, cancers, and death. Given the gravity of these adverse effects, heavy metal contamination poses substantial environmental and public health risks that must be addressed through proper monitoring, remediation, and prevention efforts. Tackling soil pollution will be critical for achieving sustainability in agricultural systems and protecting human populations from toxic exposures.

The rivers and lakes in the coastal area of the Yellow River Wetland Reserve are gradually polluted with the continuous development of industrialization. A large amount of industrial wastewater discharge has caused the water quality in this area to become worse and worse (Liu et al., 2023; Zhao et al., 2023). Ammonia nitrogen, petroleum and total phosphorus are the main pollutants affecting water quality (Chen, Yao & Zhong, 2022; Zhu, Wu & Xu, 2023).

In the Lingbao, Jiyuan, Mengzhou and Mengjin sections of the Yellow River Wetland Reserve, small family-style and workshop-style production factories are scattered, such as prefabrication factories, furniture factories, plastic film factories, and waste incineration plants. It is difficult to control the “three wastes”. Wastewater, waste residue, and exhaust gas impose certain impacts on surrounding soil environments, though currently limited in scope.

(2) The vigorous development of modern planting in rural areas has also contributed to a certain amount of pollution. According to the investigation results of the commonly used fertilizers in Henan by relevant scholars of the Henan Cultivation Soil and Fertilizer Testing Center (Wei et al., 2023; Zhang et al., 2023b), the heavy metals that exceeded the standard in the organic fertilizer samples were: As (0.034–46.68 mg·kg−1, exceeding the standard rate of 5.9%), Pb (0.013 ~ 124.15 mg·kg−1, exceeding the standard rate of 0.9%), Cd (0 ~ 21.63 mg·kg−1, exceeding the standard rate of 4.1%), Hg (0.011 ~ 137 mg·kg−1, exceeding the standard rate of 7.8%), This is because organic wastes such as sludge and domestic garbage rich in heavy metal elements are used as raw materials in the production of organic fertilizers, so that the heavy metals in the fertilizer are at a high value. Unreasonable use of feed pesticides and fertilizers, direct discharge of domestic garbage and wastewater may cause heavy metal pollution.

China’s agricultural sector applies copious fertilizer doses to pursue economic gains, accumulating in soils and eliciting pollution (Zhang et al., 2023a). Our survey of the Yellow River Wetland Reserve (Henan section) found the predominant crops-wheat, vegetables and corn-are all highly fertilizer-dependent. Approximately 1,000 kg of organic fertilizer is applied per 667 m2 annually for wheat, corn and other grains, while vegetables receive roughly 600 kg of organic fertilizer per 667 m2 per year (Li et al., 2022; Zhang et al., 2020), lead to the accumulation of heavy metals in the soil. The application of pesticides and fertilizers on cultivated land will inevitably impact the soil content of these heavy metals.

Conclusions and limitations

Conclusions

In this study, 706 soil samples from the Yellow River Wetland Reserve in Henan were used as materials, and the following conclusions were drawn through the analysis and evaluation of the contents of six heavy metals:

(1) The Nemerow comprehensive pollution index of the soil in the Yellow River Wetland Reserve is 0.42, ranging from 0.17 to 2.38, which is at the level I level of safety and pollution, and there is no risk of pollution.

(2) There is a small amount of point-like pollution in the Yellow River Wetland Reserve, distributed in the north in the form of patches, but the patch area is small. Distributed in Mianchi, Xin’an County and Lakeside in the middle, they are still clean (warning line). Distributed in Lingbao in the east and Jiyuan City, Mengzhou City and Mengjin in the west, as well as the Sanmenxia Reservoir, the Xiaolangdi Reservoir, the Xixiayuan Off-Season Water Diversion Project and the Yellow River River Service Control and Flood Control Project, are under slight pollution.

(3) There is a correlation between heavy metal elements, and the sources of indirect reaction heavy metals may be the same, and they are greatly affected by the same pollution source. It shows that these elements have a certain homology in the soil of the Yellow River Wetland Reserve.

(4) Correlation analysis and factor analysis showed that the soil heavy metal pollution in the Yellow River Wetland Reserve mainly came from the input of human activities, such as industrial pollution, ship sewage, breeding sewage, domestic sewage and agricultural non-point source pollution, followed by the input of natural factors.

Limitations of the study

In this study, 706 soil samples were selected as materials. Through the analysis and evaluation of the content of six heavy metals, in actual research, the selection of samples may be limited by time, funds and geographical conditions, resulting in limited sample space and cannot fully represent the entire Yellow River. Soil conditions in wetland protected areas. Numerous factors, encompassing anthropogenic activities and natural processes, influence soil heavy metal contents. Accurately disentangling the contributions of diverse factors to soil heavy metal levels proves challenging during analysis, potentially hindering an in-depth elucidation of the underpinnings of soil pollution.

For the area where the soil is still clean (warning line), attention should be paid to reducing the pollution brought by human activities, etc., and reducing the probability of developing to light pollution. However, because the wetland area is an area with frequent human activities, in the future research, we will study the influence of human factors on the wetland structure, reduce the heavy metal pollution of the Yellow River wetland, and maintain the balance of the ecological environment of the Yellow River wetland. Establish a mathematical model to predict the future trend of soil heavy metal content and assess the risk to the environment and human health. This facilitates better conservation strategies and decision-making.

In summary, analysis of heavy metal concentrations and pollution assessment of the Yellow River Wetland Reserve in Henan Province remains limited regarding data acquisition and multivariate impact evaluation. Comprehensive, in-depth research is imperative to inform sustainable protected area development with scientific underpinnings.

Supplemental Information

Supplemental Information 1 Data.

Click here for additional data file.

Additional Information and Declarations

Competing Interests

Author Contributions

Field Study Permissions

Data Availability

The authors declare that they have no conflicts of interest.

Xiaolong Chen conceived and designed the experiments, performed the experiments, analyzed the data, prepared figures and/or tables, authored or reviewed drafts of the article, and approved the final draft.

Cora Un In Wong conceived and designed the experiments, prepared figures and/or tables, and approved the final draft.

Hongfeng Zhang conceived and designed the experiments, performed the experiments, prepared figures and/or tables, authored or reviewed drafts of the article, and approved the final draft.

The following information was supplied relating to field study approvals (i.e., approving body and any reference numbers):

Henan Yellow River Wetland National Nature Reserve Jiaozuo Management Bureau

Henan Yellow River Wetland National Nature Reserve Mengjin Management Bureau

Henan Yellow River Wetland National Nature Reserve Sanmenxia Management Office.

The following information was supplied regarding data availability:

The original data is available in the Supplemental File.

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
