# Peer review of "Analysis and pollution evaluation of heavy metal content in soil of the Yellow River Wetland Reserve in Henan"

_PeerJ, doi:10.7717/peerj.16454_

## Round 0.1 · original submission · Major Revisions

Please incorporate all comments of reviewers and please submit a point-to-point rebuttal letter. The language of the paper needs serious attention to improve. Please also improve discussions about the sample and objectives of the study. The discussion section should also be improved.

Reviewer 1 ·

Basic reporting

There are some problems with the use of English language and grammar. I have indicated at few points. Authors can follow the same to update whole manuscript.
The literature and references are correct and sufficient.
Figures and tables are in proper shape and format.

Experimental design

The study followed the correct experimental design and the results are elaborative and concise.

Validity of the findings

These sections are well written expect for few language mistakes

Additional comments

The manuscript can be accepted after minor revisions.

Annotated reviews are not available for download in order to protect the identity of reviewers who chose to remain anonymous.

Reviewer 2 ·

Basic reporting

The manuscript is well-written and generally clear. However, there are a few instances where the language could be further improved for better clarity and precision. Additionally, some sentences or statements could be rephrased to enhance the flow and readability of the manuscript.
1) It would be helpful to provide a bit more detail on the specific sampling locations within the Yellow River Wetland Reserve. Are there any considerations for the selection of these points? This information could enhance the transparency of your study.
2) In introduction section avoid overly complex sentences and terminology that might be unclear to non-specialist readers.
3) try to make it more concise and explicit. What are the specific goals or objectives you aim to achieve with this research?
4) While you mention that there is limited research on soil heavy metal content in the Yellow River Wetland Reserve, briefly explain why this gap in the literature is important. What questions or issues remain unanswered by previous studies in this area?
5) When discussing the seriousness of soil pollution, provide a few more details about the consequences of heavy metal contamination on the environment and human health.
6) Ensure that your citations follow a consistent format. Include the full names of authors, publication years, and complete references in the appropriate citation style.

Experimental design

no

Validity of the findings

no

Additional comments

no

---

## Round 0.2 · accepted · Accept

The paper has been improved after revision incorporating all of the reviewers' comments and is accepted for publication.

Reviewer 1 ·

Basic reporting

no comment

Experimental design

no comment

Validity of the findings

no comment

Additional comments

The authors have improved the manuscript. It seems up to the mark for the publication in PeerJ.

Reviewer 2 ·

Basic reporting

accepted for publication

Experimental design

no

Validity of the findings

no

Additional comments

no